# Performance management and development system in South Africa, a necessary evil: Qualitative study

**Ozoemena Joan Ibeziako** [1,2*]

1 Department of Family Medicine, Faculty of Health Sciences, University of Pretoria, Gauteng, South Africa, 2 Specialist Family Physician, Head of Unit, Region 1 NSDR Ekurhuleni Health District, Gauteng, South Africa

* ojibeziako@gmail.com

## Abstract

Performance management focused on development enhances individual and organisational performance and enables improved services. Achieving performance management objectives is vital for addressing healthcare worker shortages and ensuring equitable, quality healthcare. A weakened South African district-based primary health care system links to inadequate leadership and governance. This study aimed to explore how doctors in primary health care perceive performance management and development systems. The objectives examined what medical officers understand about it and their experiences. Emerging themes may provide insights into enhancing implementation. This study used a qualitative, interpretive, and phenomenological research design. Stratified purposive sampling based on PMDS completion and employment duration led to four focus group discussions with 17 participants. A thematic analysis was performed. The overarching theme was the Performance Management and Development System as a necessary evil, with benefits and challenges. The subthemes included understanding components, comprehending clinic systems to improve outcomes, nurturing employee-supervisor interactions, fostering performance and learning culture, and facilitating personal and professional growth. Additional subthemes included ambiguity in fairness, lack of management capacity, and need for a bottom-up approach and realistic implementation. Effective implementation of a Performance Management and Development System requires managers and supervisors to drive this process strategically. Those responsible for clinical governance should invest in personal development to understand the process and consider appropriate implementation tools.

## Introduction

Performance Management (PM) governs work organisation [1]. The Performance Management and Development System (PMDS) framework enables continuous quality services through organisational reflection, development, and interconnectedness.

provided the original author and source are credited.

**Data availability statement:** Data are available from the University of Pretoria Institutional Data Access / Ethics Committee (contact via email deepeka.behari@up.ac.za).

**Funding:** The author(s) received no specific funding for this work.

**Competing interests:** The author declares that no competing interests exist.

Leadership contributes to the health system (HS) framework by strengthening performance and outcomes [2]. Performance Management is effective when organisational performance is linked to individual staff performance [3]. It aligns with the organisational vision by monitoring overall performance and benefiting employees and employers through job satisfaction and goal commitment [3]. A well-implemented PMDS creates high performance and understanding by defining roles, communicating expectations, and establishing achievable key result areas [4].

Globally, PMDS often face several critical gaps that hinder their effectiveness. A major issue is the lack of clearly defined performance objectives, which leads to a misalignment between individual goals and organisational strategy. Research indicates that clearly defined performance metrics enhance accountability and focus [5]. Additionally, inadequate training for managers and employees reduces their understanding of and engagement with the process [6]. Many systems also lack effective feedback mechanisms and rely heavily on infrequent reviews rather than continuous dialogue [7]. Biases, such as favouritism, halo effect, or rater leniency and subjectivity in evaluations, further erode credibility and fairness [8]. Moreover, PMDS frameworks frequently fail to integrate performance assessments with professional development plans, thereby limiting growth opportunities [5]. Finally, global implementation often overlooks cultural differences, reducing the relevance and impact of the system across diverse contexts [9].

In South Africa, primary health care (PHC) is essential for delivering comprehensive, equitable, and integrated care supported by the District Health System (DHS) [10]. Transitioning from a traditional hospicentric model, South Africa's PHC re-engineering strategy emphasises a community-based approach that promotes health, prevents diseases, and addresses the social determinants of health. This strategy fosters collaboration among community members, organisations, and healthcare teams to bridge the gap between services and users [11]. However, there is still potential to enhance the performance of PHCs in the country. This has been attributed to structural deficiencies (weak DHS) and poor regulatory frameworks (lack of accountability) [11]. Leadership and governance are among the six health system building blocks of the World Health Organisation (WHO) [11], and could address the weakened DHS. The PMDS is a tool for leadership and governance which helps improve and maintain high standards [12]. In South Africa (SA), Family physicians, "a doctor working as a medical generalist in the DHS and registered as a specialist in family medicine" [13], have been entrusted with ensuring and maintaining clinical governance to strengthen the DHS. [10]

The PMDS in South Africa is guided by the 2018 Department of Public Service and Administration (DPSA) policy with determined objectives. These include ensuring employees know and understand their expected job outcomes, establishing a performance and learning culture in public service, improving service delivery, promoting employee-supervisor interactions, and identifying, managing, and promoting growth in staff by addressing each individual's developmental needs [14]. Fig. 1 summarises South Africa's PMDS structure and processes. [1,14–16].

The Family Medicine Ekurhuleni Health District develops its annual departmental performance agreement (PA) for medical officers (S1 File), guided by the National

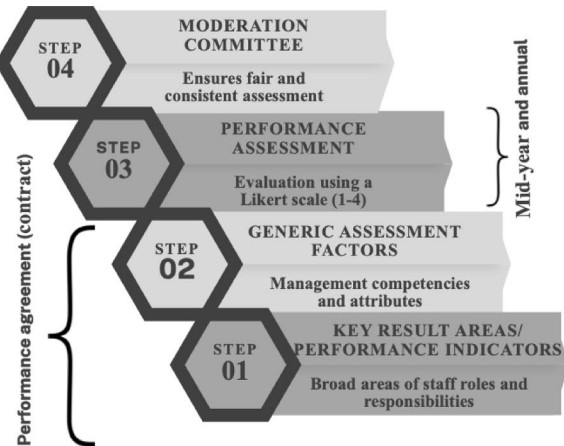

**Fig 1. South African Performance Management and Development System Framework.** This figure illustrates the key components of the South African PMDS as outlined by the DPSA policy of 2007. The framework comprises two main dimensions: Key Result Areas (KRAs) and Generic Assessment Factors (GAFs). KRAs, limited to five or six substantive outputs, describe the broad areas of staff roles and responsibilities, while GAFs encompass generic management competencies. The system employs a four-point Likert scale for assessment, conducted bi-annually, with scores influencing pay progression and potential cash rewards. A moderation committee ensures fair and consistent application of standards across departments.

Development Plan, mid-term strategic goals (MTSG), and strategic documents [15]. The Family Medicine Department of Northern Ekurhuleni Subdistrict 1 (NSDR1) has translated these outputs into a monthly practice self-assessment performance and management tool (MPST) (S2 File) for medical officers (MOs) to manage their practices and processes. The MPST measures performance through patient record audits, programme outcome analysis, educational activities, gap identification, and action planning. Performance is discussed individually, with best practices shared at meetings. This approach enables doctors to become active participants in their performance management and development.

## Problem statement

Despite over four decades of PMDS implementation in SA [1], it remains "one of the most contested systems in the South African public service" [14,17]. Literature highlights challenges around its acceptance, unfairness, weak accountability, favouritism, and overemphasis on financial incentives undermining developmental goals [17,18].

Doctors in middle management services participate in the PMDS. Given the negative discourse surrounding PMDS and the need to strengthen DHS, this study explored whether MPST influences how PHC doctors experience PMDS and its potential impact on DHS. No similar studies have been conducted in this context. The objectives examined how MOs - "a doctor employed as a medical generalist in the DHS and not registered as a specialist in family medicine" [13] – in NSDR1 understand and experience PMDS. This study could reveal the extent of challenges in this professional group and crystallise the benefits for effective PMDS implementation in PHC settings.

## Materials and Methods

### Study design

A qualitative interpretive phenomenological research design was appropriate for this study to understand the essence and meaning of the lived PMDS experiences among PHC doctors [19,20]. This design is opposed to grounded theory, which seeks to generate theory grounded in real-world observations or in-depth descriptions, as in the case study. Participants,

MOs who experienced the phenomenon under study (PMDS assessments), were the units of analysis. The researcher applied Heideggerian phenomenology to emphasise that meaning comes from interpreting participants' daily experiences [19,20]. The researcher was the primary data collection and analysis instrument, employing an inductive strategy to build meanings from the participants' direct experiences [19–21].

### Research setting

This study was conducted in NSDR1, which includes Tembisa, Kempton Park, and Northern Boksburg, in Gauteng Province, South Africa. Twenty government clinics – two community health centres (CHCs) and 18 primary healthcare clinics (PHCs) – are supervised by two Specialist Family Physicians (SFPs) based at each CHC. Medical officers are deployed to the PHC (their practice) and multiple MOs at the CHC (joint practice). All clinics provide comprehensive primary health care services. Primary healthcare facilities, including PHCs and CHCs, serve as gatekeepers to higher levels of care (hospitals) and are, therefore, the first point of contact between the community and healthcare [22]. The organisation of the PHC and CHC is depicted in Fig. 2 [22].

This figure illustrates the key components of PHCs and CHCs, including their role as gatekeepers, the Ideal Clinic Framework, Integrated Clinical Services Management streams, and the multidisciplinary workforce. It highlights the differences between PHCs and CHCs (provides 24-hour emergency services and additional 24-hour Midwife Obstetric Unit (MOU) services.

Doctors working at the CHC and PHC include various grades of MOs (Grades I to III), community service medical officers (CMOs), and medical interns who rotate every two months during their Family Medicine rotation at the PHC and CHC. The MO consults all patients referred by nurses and refers difficult cases to the SFP or hospital when necessary. The expected role of the MO mimics that of the SFP – a clinician, clinical trainer of other junior doctors, nurses, and community health workers; some level of clinical governance; and a team player in community-oriented primary care (COPC) [13]. The doctor's contract comprises output activities and indicators of these expected roles and responsibilities aligned with departmental strategic goals.

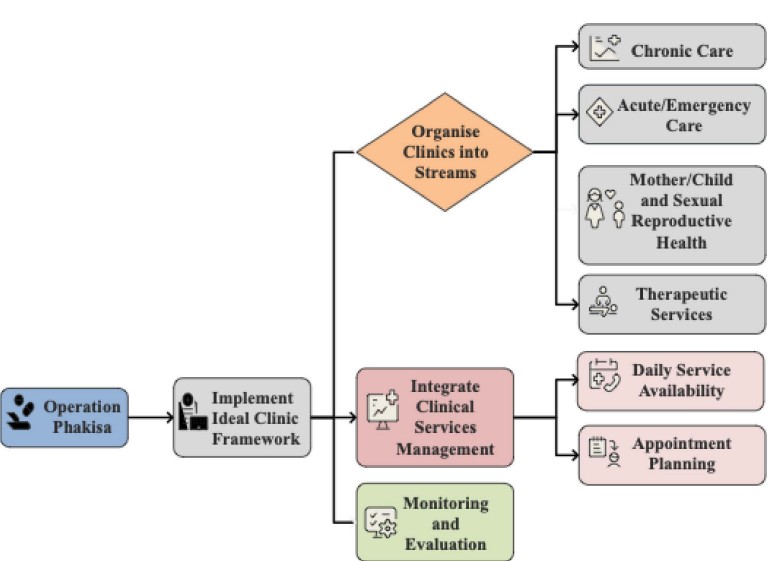

**Fig. 2. Organisational structure and functioning of primary healthcare facilities in South Africa.**

## Population and sampling

During the study period, the staff complement of doctors in NSDR1 included 7 CMOs and 14 full-time MOS. Seventeen out of the 21 MOs working at NSDR1 who met the selection criteria participated in the study. The inclusion criteria were completion of the 2022−23 annual PMDS and at least one year of employment in the unit, the minimum period to participate in the mid-year, and the annual PMDS.

Stratified purposeful sampling was then performed. Stratification by duration of employment in the unit – one year, one to two years, two to three years, three years, and above – could reveal important patterns and insights. Participants were contacted telephonically. Four MOs did not fulfil the selection criteria because they had been employed for less than one year and had only participated in the mid-year PMDS assessment. Ethical approval and permission to collect data were obtained prior to data collection.

Information leaflets and consent forms were forwarded to all MOs (17) who met the inclusion criteria and provided verbal consent. The participants signed a written consent form after perusing the information leaflet and before data collection. The total number of participants in the focus group discussions (FGDs) was six (FGD1), five (FGD2), two (FGD3), and four (FGD4). The endpoint for FGDs was determined by theoretical saturation [20,23]. Focus Group Discussions were held between October and December 2023.

## Data collection

A semi-structured interview guide (S3 File) was appropriate for exploring participants' lived experiences. The guide was piloted and validated through conducting individual interviews. No changes were made to the wording after piloting was completed, confirming its accuracy and clarity. The results of individual interviews were not included in the analysis.

Using a semi-structured interview guide, the researcher held FGDs at times convenient for the participants and in a comfortable venue. The aim of this research was reiterated for clarity. The researcher encouraged the participants to express themselves freely and sincerely, as each participant shared their experiences. Participants were aware of the importance of respecting one another. The questions were not asked in sequence but were shaped by issues raised in the prior discussion. The researcher is one of the SFPs in the unit. Although no power dynamics emerged, a research assistant was trained as a substitute investigator. To mitigate bias, it was ensured that participants had completed the 2022−23 PMDS assessment with their respective supervisors before data collection began. Data was collected until saturation was reached with the final group discussion [20,23].

## Data analysis

The researcher transcribed the FGDs verbatim using the Descript AI software and listened to the audio repeatedly while correcting the transcripts for authenticity. Thematic analysis was performed using ATLAS.ti software, which the researcher was familiar with and had easy access to. The process began with repeatedly listening and reviewing the transcript for corrections and meaning, described as "getting the sense of the whole" [19]. The transcripts were imported into ATLAS.ti where recurring words, phrases or sentences describing participants' experiences were coded as the first step [19,23,24]. This has been described as the "meaning unit" [19]. Generated codes (meaning units) were categorised as the second step [19], sub-themes, and themes emerged as the final step [19,23,24]. The researcher's familiarity with thematic analysis and peer reviews validated the codes and themes generated.

## Ethical considerations

The National Health Research Database (GP_202309_001) and the Ekurhuleni Ethics Committee (10/07/2023/01) granted permission for this study. The University of Pretoria Health Sciences Ethics Committee granted ethical clearance to conduct this study (No. 458/2023).

## Trustworthiness

Trustworthiness refers to qualitative research's "quality, authenticity, and truthfulness" [25] Lincoln and Guba described credibility, dependability, conformability and transferability as four pillars of trustworthiness [26]. *Credibility* refers to the congruence of the findings with participants' lived experiences [25,27,28]. The researcher achieved this through triangulation, using multiple information sources, such as repeated reading of transcripts and member checking of data and its interpretation with participants to establish identifiable patterns. *Transferability* concerns the applicability of the findings to a new context [25,27,28]. This was achieved by providing a detailed description of the participants and the research process (study context, sampling, demographics, selection criteria, interview guide, and analytical steps), and data collection continued until saturation [20,23] By the end of the fourth FGD, data saturation was deemed attained because FGDs, transcriptions, and analysis ran concurrently, and no new information or themes emerged. *Dependability,* which looks at consistency, was achieved by peer debriefing and maintaining the researcher's experience and interpretations (bracketing) at bay [25,27,28]. *Conformability* refers to neutrality that ensures that the findings are grounded in the data [25,27,28]. This was ensured by maintaining a tight audit trail for data collection, analysis, and interpretation.

## Reflexibility

The researcher, an SFP at NSDR1, had experience in qualitative research and participates in PMDS implementation without vested interests. Given the efforts of human resources to ensure annual PMDS completion, the researcher critically examined its benefits and limitations. The researcher engaged in self-reflection regarding potential biases that might influence data collection and analysis, refraining from sharing personal perceptions of the PMDS during data collection [25,27,28].

The researcher focused on the participants' lived experiences during the analysis, maintaining a non-judgmental stance. In the event of concerns regarding power dynamics, a research assistant was available as a substitute investigator. Collecting data after the 2022−23 PMDS assessment allowed participants to express themselves openly. Participants shared their experiences honestly, and the research results were provided to them for scrutiny and correction.

# Results

## Participants characteristics

Demographic non-personalised information regarding age, gender, duration of employment with the Department of Health (DoH) and Ekurhuleni Health District, and the number of years participating in PMDS cycles was collected from participants. Seventeen MOs were interviewed: community service (4 [23,5%]), full-time (13 [72.22%]), and 4 (23,5%) were male, while 13 (72.22%) were female, with a mean age of 41 years. In their first employment as independent practitioners, CMOs comprised most FGD1 participants and were the least exposed to the PMDS process (4 [23,5%]). The PMDS experience did not align with most participants' years of employment in the Doh. Table 1 presents the participants' de-identified demographic characteristics.

## Themes and sub-themes.

The themes and sub-themes are illustrated in Fig 3.

PMDS, seen as a necessary evil, serves as the overarching theme, with themes and subthemes highlighting an unexpected number of benefits and some challenges.

**1. PMDS, a necessary evil:** The participants demonstrated profound knowledge and understanding of PMDS. The insightful theme articulated by the focus group of experienced participants, FGD2, labelled PMDS as "a necessary evil." Despite recognising its challenges, participants effectively delineated its various components, demonstrating their understanding of its purpose and extensive benefits, which were unexpected. They recognised that PMDS, while often uncomfortable and demanding, is part of the job and plays a critical role in pushing employees beyond their comfort zones.

**Table 1. Table on participants' de-identified characteristics.**

| GROUP | MEAN AGE | MEAN EMPLOYMENT DURATION (*GDoH) (YEARS) | MEAN EMPLOY-MENT DURATION (**NSDR1) (YEARS) | MEAN PMDS CYCLES IN (*GDoH) (YEARS) |
|---|---|---|---|---|
| FGD1 (6 participants) | 30 | 3.6 | 1 | 1 |
| FGD2 (5 participants) | 59.2 | 16.6 | 14.2 | 6 |
| FGD3 (2 participants) | 31 | 5.5 | 2 | 2 |
| FGD4 (4 participants) | 40.25 | 7.75 | 3 | 3 |

*GDoH, Gauteng Department of Health
**NSDR1, Family Medicine Department Northern Subdistrict 1 Ekurhuleni Health

When executed properly, this process yields significant advantages for both employees and organisations. PMDS may be viewed as a challenging process, but its successful implementation can lead to substantial growth and development for all parties involved. Fig. 4 illustrates the contributions of the four FGDs to these themes and sub-themes.

This figure illustrates the distribution of themes and subthemes across the focus group discussions (FGD) – FGD1, FGD2, FGD3, and FGD4. Notably, FGD2 characterises the PMDS as a necessary evil, whereas FGD1 appears to highlight more challenges than others.

The challenges predominantly identified in FGD1 may suggest a unique experience, distinct from other MOs who may have developed strategies to address PMDS challenges over time. The advantages of PMDS were clearly articulated across all groups, although FGD2 demonstrated a more profound comprehension of the clinic system.

**Benefits**

**1.1.1. PMDS's multi-dimensionality.** *a. Specifies expectations, guides, a tool to identify gaps, and creates opportunities to excel.* Participants from all FGDs highlighted that the contract within the PMDS is a tool that guides, assesses, and measures employee compliance or alignment with expected outputs. The contract outlines the output activities and indicators of clinical care and governance, training of peers and other HCWs, MOs' participation in continuous professional development, and quality improvement expected of PHC doctors (SI and II). It has become a guide on job expectations and a tool to identify gaps or non-performance, especially for doctors who are new to primary health care and young, and to share experiences and good practice. Participants highlighted the importance of guiding a doctor's job expectations in the primary care setting, which tends to present a novel work environment because medical training is mostly hospicentric.

*"Maybe for us it's obvious, our roles because we have been for a long time in this business. But for the new doctors, the person that comes first to work in the clinic, to work in primary care, they won't know if they don't have something to guide them on what to do."(FGD2)*

As a tool and guide, the PMDS has output activities regarding what patient care entails in practice. It allows doctors to strive for and attain the expected clinical targets and standards of care.

*"It helped me to, in terms of how I should see my patients. I know I need to do my annual eye exam, foot exam, that's how they should be seen; that's the standard that you are setting. It gives you a sort of a guideline. I think it helped me in that sense to have a PMDS and not feel lost in the system as well."* (FGD1)

**Fig 3. Themes and sub-themes.**

If correctly implemented, the PMDS should reflect the reality in the facility and direct monitoring and evaluation to improve and maintain the standard of care.

*"I think it is a good assessment tool for public service if it is done correctly. It gives a picture of the facility and highlights areas where you are doing well and can help identify areas that need improvement."* (FGD1)

Additionally, the participants were able to pinpoint gaps in the quality of the data at the facility.

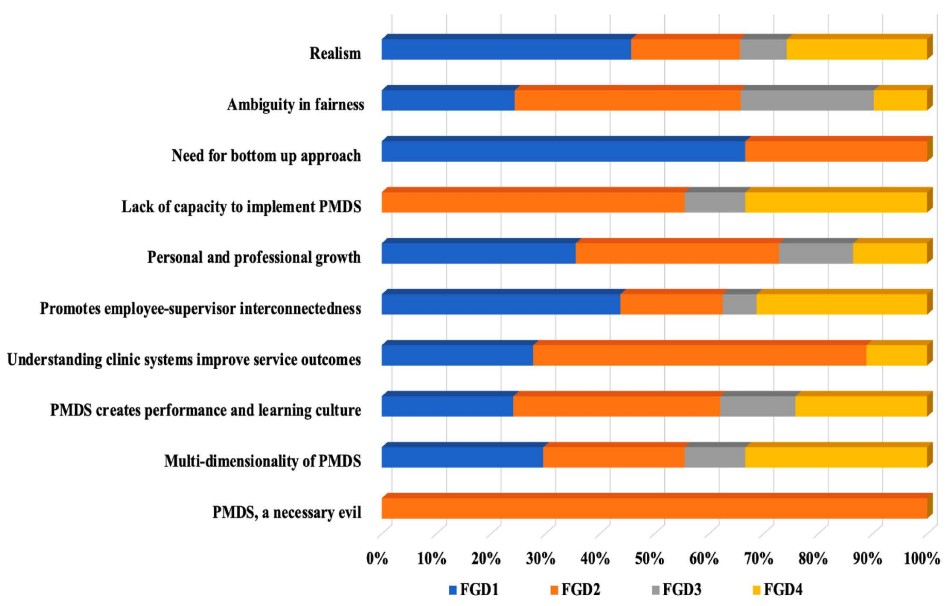

**Fig. 4. Themes and sub-themes across focus group discussions.**

*"I think that maybe the PMDS is good for us to recognise that there is a big mistake, a problem in the data."* (FGD2)

**b. Measures work alignment to expected outputs and appreciates hard work.** The PMDS assesses and *measures employee alignment with expected outputs*, *appreciates hard work*, and improves patient care quality. The MPST includes output indicators with set targets for each quarter. These targets guide doctors in gauging what needs to be done, how, and when. This tool serves as an efficient and effective mechanism for monitoring, evaluation, and implementation at the operational level. Consequently, it yields *mutual* benefits for both patients and employees.

*"...It's lovely tool (MPST). It keeps us in line. It encourages us...to do better; to do more; be reviewing and be checking what we've done, if it has benefited us in the way that we wanted to benefit. Again, if there are gaps…, keeping us abreast with our gaps. Where were your challenges? What have you done to try and mitigate those challenges? It's a tool that is guiding…where we are going in terms of how we are performing, trying to close all the other gaps that we sometimes encounter in our workplaces."* (FGD4)

All participants clearly expressed the linearity between meeting expected outcomes and cash reward according to the DPSA policy, which is a good reason to ensure that contract activities align with the expected outcome. For participants, this meant working in a directed and effective way and sometimes working under pressure to get things right. Some participants also expressed that it is an assurance that the work was not done in vain. In addition, appreciating staff efforts motivates and boosts job satisfaction. The PMDS becomes a good performance measurement if it is correctly implemented.

*"...linking it up with the contract, we have certain specifications on what we are expected to do. The PMDS is just sort of a measure to make sure that we are in line with that so that we are able to identify it (gaps) or not, then we can find ways to sort of improve on that. And if you're performing, …get compensated for the performance."* (FGD3)

*"...my understanding with regard to PMDS is that it is a tool that is used by the employer to try and assess performance of all the employees. But before that is done, there's always a baseline that is given in terms of the duties that one has to fulfil. Based on that, you check yourself more, you weigh yourself according to what is put for your baseline and what you have actually achieved at the end of the day. So just to encourage people to perform, that's number one. But also is a means to ensure that you appreciate those that have performed beyond what is just given. It's more like an encouragement because at the end of the day, it's got some monetary, …awards or rewards that will actually motivate employees to always go above and beyond what they're supposed to give in order to achieve the goals?"* (FGD4)

**1.1.2. Performance and learning culture.** ***a. Self-empowerment through reflections.*** For junior doctors unfamiliar with PMDS, the associated contract activities and expected outputs posed a substantial workload, reshaping their experience in clinical practice. Engaging in this process marked a shift toward *becoming reflective practitioners* – an endeavour that demands *self-introspection*, a rejection of *complacency*, and a *commitment to ongoing learning and improvement*. Although reflection in this context remains informal, it is deeply embedded in the ethos of Family Medicine. Here, reflective practice is not just encouraged, but is essential for clinical care, experiential learning, and the development of research grounded in day-to-day patient interactions.

*"…It has helped me understand the way I work and revealed areas that I needed to improve on, like being more organised and consistent. I've also had to work under pressure and put in extra effort, take work home, interpret and make sense of a lot of new information; that was a stretch for me but I tried my best to get things right so I guess I am open to continuous learning and new challenges. …The assessment exposed me to a lot of things, so I've had to step up, empower myself through learning and adjust quickly. Work became less stressful, and I've become more comfortable and efficient."*(FGD1)

*"In terms of self-introspection, how do you do a holistic approach to your patients?..."*(FGD2)

*"The one is that I think it stops people from being complacent. Because I think when you don't assess people, then people can just do, like not even the daily amount. They'll just do whatever …as long as they get paid. So, I think it picks up on that."*(FGD3)

Participation in the PMDS meant examining what and how they did their work and striving to give their best. The result is a continued willingness to learn, get out of one's comfort zone, improve oneself, and work.

*"…I would say, at first, I was petrified because I come from… setting, and it was only one department I was assisting. Now it was like all the departments.…When I arrived, the very first time, it was like more serious you know…As time goes on,…I get to know that I learn from it, and I got to know that I have duties, I have to follow the job description… To know what's happening in your clinic…Because if it was not for the PMDS,…I'll probably not be involved with certain things like knowing what's happening in data room, and the statistics and how they do things."*(FGD4)

***b. Build collaborations through relationships to improve patient care and services***

Learning is enhanced in collaborative environments. All participants emphasised the importance of teamwork and building relationships, expressed as bonds. Synergy between doctors and clinic teams enhanced intra-facility communication and referrals. Other aspects of building collaborations are peer support and benchmarking through sharing positive PMDS experiences.

*"..I was talking to an old lady..two weeks ago..She had some social issue, husband, fighting, she's staying in their daughter's house. Then I said, do you think you need a psychologist? She said, thanks, I need one. So, let us not be afraid to refer….You are not working alone. As a doctor in the clinic, there are other staff. There are social workers. I*

was happy when I was working with…, the podiatrist,…the dietician…With the podiatrist...I learnt how they make the patient walk.. Even the psychologist, when you read the notes…- she writes nice notes -, you see everything…This I missed a little bit in my books…That's collaborative effort." (FGD2)

*Holistic and high-quality patient care and services* can only be realised through professionals working together to improve health outcomes through *shared ideas and knowledge*. This includes working in teams and harnessing knowledge and skills from colleagues and other professionals, which provides positive performance and a learning culture. The participants understood how each team within the clinic worked and learned new skills from them.

*"I also realised that it takes teamwork to complete PMDS because everyone in the clinic contributes to the statistics in one way or the other. I've had to work with different teams in the clinic like the WBOT, ANC and MOU team, TB team… And with each team understanding how they work, e.g., TIER.NET; HAST; how we are doing as a clinic; what gaps are there, and ways in which we can improve, e.g., ESMOE drills and personally also learning new skills."(FGD1)*

**1.1.3. Understanding clinic systems improves service outcomes. a. Being part of the whole.** All participants alluded to seeing the bigger picture of the whole facility and realising that contributing to it helps them appreciate their work. In different ways, participants expressed feeling that their medical training and work were focused on clinical stuff. However, with PMDS, they recognised that patient care works through a system that needs to be in place and functional, such as ensuring the availability of medical stock in the pharmacy and, when there are stock-outs, what processes should be followed. By working with clinic teams, one is also informed about what is happening in the clinic's other services.

*"…it (PMDS) helps me to see the whole clinic or where I'm working in a bigger picture; that this is going on, this is happening there. You have a better understanding of what is going on in the whole facility.. It helps you to understand the institution as a whole, and then it gets much easier for you to work when you know that, when I need this, I need to go there… How is the service going..."* (FGD3)

For the participants, being part of a whole also meant experiencing firsthand how clinical activities at the coalface contributed to global health data, which inspired and appreciated their work.

*"…It helped me realise that I can contribute to the statistics and that all we do contributes to us as a country, meeting targets set out in the sustainable development goals. Realising this contribution made me more appreciative of my work and its contribution."(FGD1)*

**b. Advocacy.** Advocacy becomes a consequence of acknowledging one's role in a system. PMDS-experienced participants recognised that their roles as *clinical trainer*s of junior doctors, nurses, and other healthcare professionals, *patient advocates*, and *resource managers* impact the health system and quality of services. Educating other team members empowers them and creates a joint vision in clinical care using limited resources.

…*"And your role as well, to be the advocate of government's resources."(FGD2)*

*"I now make Wednesday topics that is about medicines that usually we prescribe. I start with Lanzoloc. I already did prednisone. Because I have one or two sisters who order prednisone for flu or for arthritis… I said, every Wednesday we're gonna talk about different medications that are wrongly prescribed."(FGD2)*

Exercising the patient advocacy role meant working professionally and communicating appropriately; the doctor ensured that patients accessed the appropriate level of care and awakened responsibility and accountability among other clinicians and colleagues.

"PMDS has taught me something,..the clerking of a patient.…*I sent a patient with COPD, a very severe COPD to General OPD. This patient was just seen and given Asthavent and was not given even any date. I phoned…, I said, 'ah, this is unacceptable.' The sister said, 'no, no, the doctor who saw this patient is usually very thorough'…I said,... 'I want something written down, and I want something to come back.' Hey, he (the patient) destroyed his life and everything…But now, let's just prolong his life a little bit, with dignity and quality of life.' When I said that, it's like, 'who are you? Where are you phoning from?'...'Okay, doctor, I will ask the doctor to see the patient'… They saw the patient again. And the patient came back. Now he comes to collect the Symbicort. He is improved, guys. He's improved."* (FGD2)

**c. Improved patient experience of care through effective systems.** Participants indicated the usefulness of clinical audits included in the MPST with varied examples, prompting the active use of internal resources, interconnectedness, and effective strategies, such as developing standard operating procedures. Good patient care, especially chronic care, is achieved and sustained through the on-going monitoring of patient progress. Improved professional skills through upskilling oneself and thinking outside the box positively affect the patient care experience.

*"…I was looking at the fact that, during COVID, a lot of patients were taken out of the system and either put on CCMDD without even seeing doctors. Those patients had never seen doctors. So you find them, like, once in a while…What the PMDS has done, it has brought into my thinking, I've had to think outside the box of what to do with those patients… What I then devised was a plan,…I go into WBOT and actively call…all those patients and see what I can do for them. And because of that, I've seen so much improvement…"* (FGD2)

*"... it also pointed out the shortfalls. If we didn't do a PMDS, I wouldn't know where my shortfalls are. For me, the referral system was heavily short-falled. It took a while, but eventually,..having to tell the sisters to call even if I am not duty on that day. Call me and discuss a patient that you want to refer. Cause sometimes you find that it doesn't need to be referred, you can actually manage it yourself."*(FGD1)

Participants articulated that improving the patient's care experience entails paying attention to little things in clinical practice that might not require much effort, but are meaningful for the patient. The biopsychosocial approach – taking time to talk to patients about their psychosocial situations, screening and health promotion, and educating patients on lifestyle changes–empowers them to be responsible for their health. Thus, patient satisfaction was assured, and participants learned and practised patient-centred care.

*"...I think the greatest win for me this year was realising how merely screening for mental health, which is often the big elephant in the room, changes a lot of things. I have definitely shifted my style of consultation and have been appreciative of having witnessed its impact on a patient's life, their attitude towards treatment, and their response to treatment."* (FGD1)

Improved patient care experience was described as individualising patient care to address differing needs from a biopsychosocial approach and developing clinical care tools, such as diabetic diet charts, to enhance patient care. Participants became more cognizant of the social determinants of health and how they could impact patients' adherence to treatment goals, if not considered.

*"…if we talk about chronic patients,…the first thing that it (PMDS) has improved me is to individualise every patient... Even the ones that are doing well, I must take them as individuals. I… write SOPs. I've designed my own charts for my diabetics…I have got a way of talking to my patients. The approach is not always the same with every patient. Psychosocial has been my main thing that I use with my patients - the state of their poverty, the state of their wealth, the state of their families, and economics. Without PMDS, I don't think I would have looked. I would just have gone*

*through a patient, and as long as the next is coming. Now, I don't look at the patient as a burden, as someone who is kind of, oh, what is he going to say today?"* (FGD2)

**1.1.4. Promoting employee-supervisor interactions and interconnectedness.** The participants agreed that PMDS *makes sense* and is possible if the supervisor provides training, guidance, and tools to make it understandable and meaningful. *Training* on the MPST at the beginning of the year, especially for junior or new doctors, guided practice, and the *feedback* loop enhanced *employee-supervisor interconnectedness*.

*"…while I was working on my first PMDS…I was still trying to figure things out. I only realised what was expected of me when I was handing in my first PMDS assessment. There were things I needed to correct and data that I had to inter- pret differently. That's when I was guided through the process and had a better understanding. The meeting we had about PMDS also cleared a lot of things."* (FGD1)

*"To me, PMDS, as a junior medical officer,…when I…was introduced to it, it was something new. So, it was a lot of work…But with more understanding and being in the system, and understanding how the system works, then it started making sense."* (FGD4)

An employee wants to improve and make a difference if the goal is clear and the process makes sense. Training and follow-up meetings also provided a platform for benchmarking and learning from other colleagues.

*"I feel like PMDS does make sense as a doctor. I am not sure if being a community service doctor really entails PMDS. But definitely, after that (meetings), it makes more sense because, at the same time, you are improving yourself as well as the standards of the clinic, and at the end of the day, all we want is better patient care."*(FGD1)

Participants in FGD2 and FGD4 described non-monetary rewards, such as certificates of appreciation, feedback, and patient appreciation, as boosting staff morale. More value was attached to professional prestige and growth, patient respect and feedback, and patient satisfaction with services received than to monetary compensation.

*"…It's not about the money. And I've always said…it's not about the money. It's about how I manage my patients. And the feedback that I get…I feel so good when I get feedback from my patients…"* (FGD2)

Participants in FGD4 also acknowledged that non-monetary forms of appreciating employees' commitment and dedica- tion will drive ethical and professional practice.

*"…I think it (PMDS) should be..put up nicely because it sometimes tends to promise what…the department cannot really do for the employees. It just should be used as a tool and not promise any type of reward other than certificates, not the monetary one. Because that one..makes people..cut corners in order to achieve your best scoring because… they want the money. Rather than…being given money, can we then get.... certificates where you're appreciated for the work that has been done? …It will take away some of the issues that we've…observed…what people will do to try and get money… when they have not done justice in terms of what their job description is."* (FGD4)

**1.1.5. Personal and Professional Growth. a. Role identification, sense of direction, meaning and purpose.** Personal and professional growth was welcomed as an uncomfortable aspect of growth triggered by PMDS. These were driven by *role identification*, a *sense of direction*, and finding *the meaning and purpose of work*. Although PMDS with the MPST put much pressure on work due to the many new things one had to learn, it changed one's attitude toward work and made work meaningful, motivating, and satisfying. Participants allude to learning to plan, prioritise and set work goals and identify and develop technical skills such as becoming computer literate.

*"…It helped me identify that I…have a bigger role than I initially thought. Because with the previous place where I worked, I used to just…be a clinician on the ground, do my work, see patients, and go…As soon as we got here and did the PMDS, then we started recognising that….there are some things that needs to be done in the clinic and that progress requires us (doctors) to…do them and plan..and move them forward..Like..planning on drills,..educating other team members, that rely on us…We saw that we are the ones who actually can do it. …Things like implementing BMI, which we didn't have before, it was a matter of..talking to the staff members..to say that the equipment is there, so let us get working so we can..start implementing and doing better on PMDS."* (FGD3)

*…"yes, it has helped in terms of putting priorities at work…according to how your performance was on the previous assessment in order to cover for the gaps…"* (FGD4)

*"PMDS actually sets goals for you…. it helps you plan your daily work on a daily basis and then achieve what you need to achieve."* (FGD4)

*"…it has helped me, where you are limited,… and then what you need to really improve on things like…, being computer literate..."* (FGD4)

Participants in each FGD explained that identifying one's bigger role as a doctor and planning, supervising, and implementing changes with team effort becomes a game-changer in approach and reason to work. Although daunting to a young doctor, with support, it drives creativity and growth.

*"This one (PMDS) is quite new. How do I even meet people who are senior to me and give feedback and say, guys, here we are not doing well. So... (PMDS) also helped with being in the front line and supervising everyone, and that's one thing that I was lacking - being a supervisor to interns, nurses,.. guiding everyone. And I'm still young…it's one of my roles, my duties, and I really need to help all these other healthcare workers."* (FGD4)

In another FGD, participants described negative experiences of lacking meaning in work as boredom, disappointment, and feeling bad about going to work.

*"… the fact that if you go to the clinic - and I have had that experience from years before - if you go to the clinic to kill the queue, to finish a thing, you feel so bad going to work like that. You feel bored, you feel disappointed, and you don't like your work, you want to do other things..."* (FGD2)

**b. Broadened horizon of care.** Reciprocity between personal and professional growth is evident through the influence of PMDS, which fosters *greater humanity, innovation, and enhanced professional skills,* ultimately leading to *holistic patient care*. Participants in FGD2 highlighted the emergence of *professional prestige and respect* from this integrated growth. Their broadened care horizons encompassed deeper empathy, recognising patients as whole persons, actively seeking health information, and adopting innovative strategies to engage patients and enhance care. Additionally, heightened awareness of their agentic roles and advocacy efforts has further expanded their understanding and practice of care.

*"PMDS has improved me as a human being as well, to be empathetic,… not sympathetic…We must never be sympathetic to our patients…"* (FGD2)

*"…you find yourself looking for information, and you find a lot of things that you would never be informed because it was maybe discussed at another level…For instance, …there is a program that is 'welcome back patients to care'. That is, never reject the patients... The patients are our team. Just welcome them back…. 'Thank you for coming;' 'I'm happy that you've come;' 'You didn't come, let's see the reasons.' …It's changing the approach to the patient…I learned that because I was looking for how to improve this thing (defaulters). Because we were having a lot of defaulters. So, it makes you look for information to improve your treatment, your management as well."* (FGD2)

Participants increasingly recognised and communicated the significance of patient education that takes into account the patient's socioeconomic status, cultural context, and health belief systems.

*"…talk about the socioeconomic of a patient, our black patients, they like pap.. If you say reduce your pap, it means, if it was like this, it must be like this…."* (FGD2)

As alluded to in all FGDS, one not only becomes a manager but also acquires *visionary leadership* by going ahead of the team, foreseeing, planning, influencing, and effecting changes. Performing the PMDS helped participants understand their roles and responsibilities, which extended beyond the doctor's consultation room. It broadened the horizons of care and practice.

**FGD1:** *"It has helped me see things beyond just being a clinician. I've had to learn about the social issues people face and understand the impact they have on patients' health and wellbeing and correlate it with the statistics we have on different outcomes. I've had to learn about leadership,… different factors that contribute to service delivery and my role in all of that. I also identified things I enjoyed more at work like women reproductive health and PMTCT (prevention of mother-child transmission). Having to do different things broadened my understanding of my role as a doctor at a PHC level."*

**Challenges**

Participants recognised that, despite PMDS's multiple benefits, there are challenges to its efficient and effective implementation that managers could address.

**1.2.1. Lack of capacity.** Managers in some units appear to lack the capacity to implement PMDS because of a lack of understanding and ability to do so. As participants in two FGDs alluded to, sometimes PMDS scores are 'cooked' or made up solely to get monetary rewards. The consequence could be that PMDS does not reflect employees' actual performance, but becomes mere compliance.

*"As a junior medical officer,… I never even thought of completing PMDS. But where I was at the time, we had an option of not completing. It means that our managers at that point in time did not understand what the purpose of PMDS was. And then from there, as I grew.. in the career,..they would complete it for us and just let us sign…It was more of a tool just to comply with what the employers wanted..We just go and sign. No proofs..They'll just give you a '3', and you'll be content with some satisfaction because you do not want to go beyond what you are supposed to do…I think most of the attitudes developed from that going forward."* (FGD4)

Participants expressed that managers' lack of capacity to manage PMDS is evident in employees' scores beyond objective performance, which are misaligned with the clinic's performance.

*"PMDS should be a true reflection of what's happening in the clinic. There should be accountability and evidence for scores that are given, and people should have an understanding of what different scores mean. The statistics should correlate throughout the year and with any significant change, be it a drop of improvement. There should be factors identified that contributed to that, and somehow, that should make sense. What's on paper should be what's on the ground."* (FGD1)

**1.2.2. Bottom-up approach.** Participants in FGD1 questioned what informs the PA targets because they sometimes exceed the reality of a particular facility, such as its patient population or socioeconomic status. In some instances, a bottom-up approach to setting contextualised, realistic, and feasible PA standards involving employees on the ground could make targets more plausible. There is a need to align the contract with reality.

*"It is a good way, but performance assessment can't be one-size-fits-all because working conditions are not the same for everyone. For some people, given the resource constraints they face or socioeconomic factors affecting the populations they serve, it may be difficult to reach the set target, which means no matter how hard they work, they may never reach the targets and may never get remunerated."* (FGD1)

**1.2.3. Ambiguity in fairness.** Participants cited factors beyond their control, such as resource shortages (equipment and staff) and patients' non-adherence to care targets. They also noted that poor team collaboration, which is essential for quality patient care, led to low PMDS scores, despite individual efforts. Resistance to change was highlighted, emphasising that both achieving and sustaining improvements require strong, ongoing team collaboration.

*".. it makes sense why it's (PMDS) there, but I feel like for our work environment and the resources that we've been given, and then the targets that we are given, it's not fair...because…we're basically almost making a plan every day for most patients..."* (FGD3)

*"…some areas are not fully reliant on you as a clinician…It's teamwork, and sometimes you have people that no matter how much you tell them,... they still don't do it, and... you… score lower. Then it's not a reflection on you, it's a reflection on the group…"* (FGD1)

*"..it (MPST) is quite helpful….I will say it's not perfect because there are challenges. Some of the work that we have to do depends on the nurses to do their work and you have different personalities at work. There are those personalities, even if you tell them, do this, even with the manager being involved, they just refuse to do things the way…If it there was no PMDS, I don't know, how would I be working."* (FGD4)

*Power dynamics* between facility managers, clinic administrative managers, and doctors tasked with clinical governance make it challenging to bring about changes to attain set standards. All these factors create questionable *ambiguity in fairness.*

*"…For instance,… my clinic…because the manager retired,... we don't have a manager…So, we don't have good evaluations in the ideal clinic…I've been trying to help them and tell them, let's do quality meetings, but it's not in my power... I believe that doctors need to have a little bit of more power in the decisions made in the clinic. They have to consult us and ask us things... Because they do their own things, and doctors do their own thing."* (FGD2)

*Staff absenteeism and the non-replacement of retired managers create a management vacuum that increases workload.* Consequently, the clinic doctor adapts daily set work plans to compensate for the staff shortage.

*"…It's just that the situations in our clinics,… are not ideal, in the sense that it's overwhelming, the work that we do… Like we were talking about absenteeism. On a day, sometimes, you have to intervene and help out, even beyond the number of patients that you've been booked for. Because, in any case, patients have to be serviced. They are here now. And there's only one chronic sister. Such things, then you get… derailed… in whatever you want to do…."* (FGD2)

The fairness of PMDS is questioned when disparate units are evaluated using different metrics. Given that outputs reflect collective team efforts, suboptimal performance should be attributed collectively, beginning with the facility manager. However, this practice is infrequently observed, leading to staff frustration.

*"..It (PMDS) has to be taken seriously by everybody that's on the system because you are assessing me for the evaluation of the ideal clinic of my clinic. But then with the manager, she has a 5, and she makes them get the bad evaluation.. It's more their responsibility than mine…The nurse have to do this,.. and they don't..and then that affects also my evaluation. But then they are not being assessed as strictly as we are being assessed."* (FGD2)

**1.2.4 Realism.** Despite the MPST, PMDS remains stressful due to heavy demands, such as clinical audits and gathering extensive evidence for high scores, often hindered by limited data team capacity and poor frontline data collection. While initially met with resistance, some participants later recognised PMDS's value for personal and professional growth. However, many felt that the effort required to achieve high scores was exhausting, and time constraints remained a major challenge. The result is inconsistency between use and last-minute practice assessments. While most suggested quarterly reviews to ease the burden, others preferred monthly reviews for continuous improvement, highlighting the need for a bottom-up performance agreement.

*"I was not consistent with using the tool, I often left things for last minute which made data collection difficult. I think the monthly tool is a good way of keeping track of your record and having a good sample size."* (FGD1)

*"..I felt it was asking too much of me because..I had so much on my plate..I got to get this, I got to get that…..I got to photocopy and everything. It was a struggle…I really had a negative attitude towards it. Very negative attitude. But the last three years, when I started doing it quarterly… it reduced the amount of work that I was supposed to do. I now had a system that was within me on how to tackle it..It has improved me as a clinician and also as a statistician.*" (FGD2)

Participants noted that the results from practice audit using the MPST could be skewed if doctors selectively choose only well-documented patient files, highlighting the critical importance of integrity in the process.

*"..you can choose the patients that you want. The controlled diabetes, the controlled cholesterol…Some of them were wonderful… every tool has got advantages and disadvantages. In research,…..they call a gap in the research…that's the bias."* (FGD2)

*"I was going to put it in this way, that we need to change the sampling of those patients. When you get a sample, we need to know or devise means of how this thing must be done. Because, number one, if you then choose your patients because you want good marks, it's not representative of the clinic."* (FGD2)

### Summary of key findings

The study interviewed 17 medical officers, including four community service and 13 full-time practitioners, with a mean age of forty-one years and predominantly female composition. Participants viewed the PMDS as a "necessary evil," acknowledging its challenges while recognising its extensive benefits in pushing employees out of their comfort zones and yielding advantages for both individuals and organisations. The PMDS offered several benefits, including its multi-dimensionality in specifying expectations, identifying gaps, and measuring work alignment. It promoted a performance and learning culture by encouraging self-empowerment and building collaborations to improve patient care. The system also improved understanding of clinic operations, enhanced employee-supervisor interactions, and fostered personal and professional growth by helping participants identify their roles and broaden their horizons of care and practice. However, challenges in PMDS implementation in the health sector were noted, including a lack of managerial capacity for effective implementation, the need for a bottom-up approach to setting targets, ambiguity in fairness due to factors beyond individual control, and issues with the PMDS's realistic implementation at the operational level.

### Discussion

PMDS's multidimensional benefits in this study align with the DPSA's [14] goals of the PMDS. Although factors may be beyond one's control, PMDS implementation strategies could positively impact employees' experiences, job satisfaction, and development. The South African Public Service Commission (PSC) report found that an alignment between individual performance goals and departmental strategy exists, but the approach varies across departments. This variation creates a trade-off between aligning with high-level strategic goals and responding to immediate operational priorities [4].

## Training with support provides knowledge and appreciation of the PMDS

Participants expressed an in-depth understanding of the PMDS. They appreciated the unique place of having a PA or contract that outlines agreed-upon activities directed towards expected outputs. Training and guided practice enabled participants to articulate the why and purpose of PMDS, in keeping with some definitions of PM by Aguinis as a continuous process of identifying, measuring, and developing the performance of individuals and teams and aligning performance with the organisation's strategic goals [5].

The MPST guided participants on the expected output indicators. Training at the beginning of the financial year, quarterly submissions of MPST reports, and meetings to share best practices have enhanced its use. The MPST enabled the participants to self-assess, monitor, continuously correct, and implement changes. Studies have highlighted the critical role of training, which informs and empowers managers and employees, but is often lacking [29,30]. Training on the PMDS becomes effective if there is practice guidance. Lockwood argued that employees are more productive if they have the necessary knowledge, training and development to do their jobs [31]. An eight-point managerial PMDS implementation strategy described by Sambo and van der Walt still leaves a vacuum in its practical translation at the coalface [32]. The positive impact of the MPST is significant because Melnyk et al. [33] indicated that PM could adversely impact performance when misalignment exists between what the organisation wants to achieve and what is being measured. From the lessons learned from other countries, the 2018 PSC report concluded that coherent PMDS is vital. While implementation is complex and context-specific, common principles like clear standards, aligned goals, professional development, and accountability are essential for a successful implementation [4].

Participants accentuated the ongoing nature of PM, which is opposed to performance appraisal–a reductionist approach to the essence of what PMDS is supposed to achieve. Smither and London reported that managers who lack this understanding make PM a mere tick-box exercise [34]. In addition, for Aguinis, evaluations performed twice a year without constant coaching and feedback to the employee are just performance appraisals and not management [5]. The fact is that performance appraisal should form part of PM [5,35]. In the study by Govender and Bussin, performance appraisals were confused with PM, which resulted in poor strategic focus [36]. In Mboweni and Makhado, nurses only submitted appraisals for the entire quarter during the annual PMDS; there was no opportunity for feedback or continuous monitoring [37]. The Gallup Group study concluded that knowing what is expected and the necessary tools to perform the job drives employee performance [38].

Effective PMDS implementation tools depend on how seriously employees and managers use them [39]. Therefore, managers, supervisors, and employees must be trained [39–40], proactive, familiar with the PMDS policy, and creatively think about the effective implementation of PMDS.

## Opportunities for collaboration and growth drive job satisfaction and achievement of the organisation's goals

Building an employee engagement (EE) culture will leverage and drive performance [36–37]. An engaged employee works positively to change behaviour and improve operational performance. Gruman and Saks highlighted the growing popularity of EE as the key to an organisation's success [41]. According to Robinson et al., EE is a positive attitude towards the organisation and its values [42]. An engaged employee knows the business context and works with colleagues to improve job performance for the organisation's benefit. The organisation must work to develop and nurture engagement, which requires a two-way relationship between the employer and the employee. An engaged employee is more than a nice thing; it has become a must [42]. The MPST enabled employees to engage with departmental strategic goals through interprofessional collaboration and learning, resulting in professional development and personal growth.

Schaufeli et al.. explained that engaged employees can deal with demanding jobs and are connected to their work activities. In this study, although the participants experienced an initial sense of increased workload with the PMDS, its guided practice (MPST) added value, meaning, and purpose to work, triggering discomfort that resulted in professional

and personal growth [43]. This is unlike other studies in which employees developed negative feelings from favouritism, tension, competition among colleagues, a lack of meaning, and no opportunities to develop themselves [18,37,44]. PMDS is perceived as punitive [18]. The PSC 2007 toolkit for managing poor performance highlighted the primary orientation of PM as developmental [17]. The MPST facilitated staff introspection, reflection, and identification of gaps and improved professional skills, innovation, collaboration with colleagues and other professionals, learning, and growth.

The unique experience of the participants in this study was that working together among doctors and other healthcare professionals created a learning culture and improved the intra-facility referral system. The MPST provided guided self-directed learning and development, motivating participants. Robertson-Smith maintains that engaged employees feel empowered and grow in confidence in their ability to do their jobs [45]. Collaboration among teams improves health outcomes, although it requires hard work. The lack of team effort in some studies was a significant barrier to PMDS implementation [18,37,44].

The MPST, an ongoing self-monitoring tool, allows employees to measure, identify gaps, and make decisions to correct and implement changes. This is consistent with Menon's [46] description, but unlike Lutwama et al. [47]. The Gallup Group identified critical employee expectations for engagement as wanting to know their expectations and the necessary tools to perform their jobs. They want to use their skills, talents, and recognition. It was vital that they felt valued. Feedback on their performance was necessary; they wanted learning opportunities and good relationships with their co-workers [38].

Khan described that engaged employees express themselves physically, cognitively, and emotionally during role performances [48]. Some participants expressed being more humane and empathetic, incorporating behaviour change that contributed to the patient's positive care experience. Unlike this study, the lack of adequately prepared and skilled supervisors – managers merely complied, creating a vacuum of meaning and purpose for PMDS. Consequently, employees are not trained, PMDS is implemented with no goal-oriented strategies, and accountability is not enforced [18].

## Authentic leadership

Gruman and Saks linked the success and effectiveness of PM to the quality of leadership that engages, provides support, challenges, coaches, gives constructive feedback and includes employees in the organisation's activities [41]. As a clinical leader entrusted with governance, the SFP can aptly harness entrusted MOs' abilities and broaden their knowledge and practice horizons through clear role identification, support, and strategic tools. This is unlike the study by Govender and Bussin, in which a lack of management and quality leadership negatively affected performance [36]. Nxumalo et al. concluded that PMDS implementation at the local level depends on the ability of deployed managers to perform in a transforming and complex environment [18].

Insights from Jane and James Sang's study show that transactional leadership (exchanging rewards for services) and transformational leadership (inspiring employees toward a shared vision) positively impact PMS success. They emphasise that PMS should be a continuous cycle of planning, coaching, counselling, and appraisal – not just goal setting and year-end reviews [49]. Leaders act as key change agents who influence the adoption of new practices and processes. Analysing leaders' values, traits, competencies, motives, style, behaviour, and skills is essential for successful PMDS implementation [50].

Managers must develop hard skills of planning, coordinating, and monitoring and soft skills in relationships and communication that promote collaborative and shared management. The South African DPSA Leadership Development Strategic Management Framework describes how leadership behaviour should add value to management roles through emotional, social, and cultural intelligence, essential soft skills [50]. The findings in this research regarding the meaning and purpose of work agree with Lockwood's description of the benefits of a workplace culture that provides employees with a sense of meaning and purpose in their jobs [31]. Work-role fit and support connect leaders and employees to shared beliefs and to attain organisational goals.

## Challenges

Bourne et al. described employees' perception of the PMS as poor and likened it to the 'big stick' due to a lack of involvement in PA goal setting [51]. This aligns with Madlabana et al.'s emphasis on setting realistic goals with staff to maintain care standards [29]. The participants expressed their desire to be involved in setting achievable output targets. The PSC emphasises the role of senior management in establishing PM standards and transformation indicators. Poor departmental planning leads to inadequate PA and ambiguous performance standards. A bottom-up approach to goal setting without a strong top-down direction can weaken accountability and organisational performance. Effective performance management requires coherent management structures and strategic leadership. The National Development Plan advocates strengthening both top-down hierarchical oversight and bottom-up accountability, whereby citizens hold officials accountable for service delivery [14]. This requires managerial self-introspection.

Participants also emphasised other non-monetary rewards, such as certificates of appreciation. In the study by Govender and Bussin, non-monetary recognition of the employees' good work intrinsically motivates [36] and boosts employees' morale [37]. Participants referred to ambiguity in the fairness of PMDS and monetary remuneration because they were aware of other units where employee scores did not equate to deserved performance, reflecting the manager's lack of understanding of the PMDS. In addition, some employees are driven by monetary rewards [37–38], and the PMDS lacks fairness, openness, and transparency and is subjective with unclear guidelines [36–37]. It creates tension and competition among employees because rewards are not equally received, thus ruining relationships [2]. Managers are fearful of giving unfavourable feedback and low scores [18].

PMDS implementation using the MPST faced major challenges owing to workload strain, rushed data collection, and inconsistent use. Participants struggled to balance routine tasks with data collection demands and often completed assessments in the last minute. Similarly, the Mopani district study reported staff frustration due to unclear policies and a lack of feedback, although there was no formal implementation tool – only form completion without structured support [37].

Although a systematic process for randomly selecting 20 patient files was prescribed, the participants highlighted a potential lack of data integrity if doctors chose better-documented files to present more favourable evaluations. In Mopani, while file selection practices were not described, bias still occurred through favouritism and inconsistent rating practices, which were worsened by the lack of clear guidelines. Additionally, sampling practices could undermine the credibility of PMDS. In Mopani, the issue stemmed more from the superficial, one-time nature of evaluations than from selective sampling [37].

Finally, although initially resisted, PMDS was later perceived as valuable for personal and professional growth. However, the effort required remains exhausting, time constraints persist, and studies show that weak managerial support and inconsistent enforcement by supervisors often undermine its full benefits [18].

In conclusion, training and guided practice enhanced PMDS implementation, while the MPST facilitated self-assessment and improvement. Collaboration and EE positively impacted job satisfaction and goal achievement. Effective leadership, both transactional and transformational, is crucial for successful PMDS implementation. Challenges include realistic goal-setting with staff involvement, balancing top-down and bottom-up accountability, addressing fairness concerns in evaluations and rewards, and managing workload strains from data collection and assessment. PMDS is perceived as valuable for personal and professional growth, despite initial resistance. Time constraints and weak managerial support can undermine PMDS benefits. These findings provide insights into PMDS implementation in the South African public service and identify areas for improvement and future research.

## Strengths and limitations

The study's strengths include providing insights into PMDS experiences of an under-researched group of healthcare professionals in PHC settings, using an exploratory interpretive qualitative approach that allowed participants to narrate and interpret their lived experiences. The researcher employed stratified purposive sampling to select participants who

experienced the phenomenon and could provide rich, in-depth information, achieving data saturation with 17 participants across four focus group discussions. Additionally, the study introduced a monthly practice self-assessed performance and management tool, potentially influencing how this group experienced PMDS.

Limitations of the study include its focus on medical officers in one specific region, limiting generalisability to other regions, districts, or low-middle-income countries. There is potential for social desirability bias due to the researcher being one of the two supervisors conducting the focus group discussions. Employing independent researchers or neutral facilitators for future FGDs could minimise bias potential. The study did not explore how PMDS implementation can be improved, which could be an area for future investigation. Furthermore, the findings may not represent experiences of other healthcare professional groups or settings. To enhance external validity, similar studies could be conducted in different settings by developing and testing interventions based on the study's findings, which may improve PMDS implementation in PHC settings. This approach could involve engaging a larger and more diverse group of healthcare professionals.

**Implications/recommendations**

The study's findings have led to several practical steps, which are presented as implications and recommendations for policymakers and health administrators:

1.  Develop and implement an effective PMDS tool

Creating a suitable PMDS implementation tool that aligns with output indicators and provides clear guidelines for fair, transparent, and contextually relevant performance evaluations. Additionally, regular reviews will ensure continuous system improvement.

2.  Capacity building and on-going support

Providing comprehensive training for employees and managers on the use of PMDS, along with on-going support, leadership development programs, and the promotion of both transactional and transformational leadership skills.

3.  Promote reflective practice and professional growth

Encourage clinicians to engage in reflective practice to foster continuous professional development, job satisfaction, and improved clinical governance.

4.  Foster collaboration and shared learning

The strategic placement of the SFP could strengthen clinical teams and governance, create structured opportunities for inter-professional collaboration, and share learning experiences.

5.  Recognition and motivation

The implementation of non-monetary recognition initiatives to acknowledge and appreciate staff performance can boost morale, ethical practices, and job satisfaction.

6.  Employee involvement and workload management

Involve frontline staff in setting realistic performance targets and developing strategies to manage workload challenges, ensuring consistent and sustainable PMDS implementation.

**Conclusion**

These findings highlight the importance of PMDS tools with adequate training and support. Success factors include promoting growth, finding meaning and purpose in work, improved patient experience of care, and inter-professional collaboration. Specialist Family Physicians play a crucial role in clinical governance within the South African DHS and can

contribute significantly to addressing the challenges of doctor distribution. In addition, SFPs, focused on PHC and DHS, could strengthen DHS by enhancing clinical governance through interprofessional collaboration and fostering positive learning. The experiences of these professionals confirm challenges with the South African PMDS while highlighting its unexpected benefits and emphasising the need for effective implementation and clear leadership.

## Supporting information

**S1 File. Sample of the annual contract signed by each medical officer.**
(PDF)

**S2 File. Sample of the translation of the annual contract to a self-assessed management tool.**
(PDF)

**S3 File. Qualitative questionnaire.**
(PDF)

## Acknowledgments

The author expresses gratitude to the medical officers who contributed their valuable time to this study, as well as to the colleagues on the peer-review team.

## Author contributions

**Conceptualization:** Ozoemena Joan Ibeziako.

**Data curation:** Ozoemena Joan Ibeziako.

**Formal analysis:** Ozoemena Joan Ibeziako.

**Investigation:** Ozoemena Joan Ibeziako.

**Methodology:** Ozoemena Joan Ibeziako.

**Project administration:** Ozoemena Joan Ibeziako.

**Resources:** Ozoemena Joan Ibeziako.

**Software:** Ozoemena Joan Ibeziako.

**Supervision:** Ozoemena Joan Ibeziako.

**Validation:** Ozoemena Joan Ibeziako.

**Visualization:** Ozoemena Joan Ibeziako.

**Writing – original draft:** Ozoemena Joan Ibeziako.

**Writing – review & editing:** Ozoemena Joan Ibeziako.

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
