## [Decision Letter · Decision Letter 0]

Dear Dr. IBEZIAKO,

We look forward to receiving your revised manuscript.

Kind regards,

Margaret Williams, Ph.D

Academic Editor

PLOS ONE

Journal Requirements:

2. For studies involving third-party data, we encourage authors to share any data specific to their analyses that they can legally distribute. PLOS recognizes, however, that authors may be using third-party data they do not have the rights to share. When third-party data cannot be publicly shared, authors must provide all information necessary for interested researchers to apply to gain access to the data. (https://journals.plos.org/plosone/s/data-availability#loc-acceptable-data-access-restrictions) 

3. Please ensure that you refer to Figure 1 in your text as, if accepted, production will need this reference to link the reader to the figure.

**Additional Editor Comments:**

Please review the comments provided and adjust the article as requested. We look forward to receiving the corrected manuscript at your earliest convenience.

Reviewers' comments:

Reviewer's Responses to Questions

**Comments to the Author**

1. Is the manuscript technically sound, and do the data support the conclusions?

Reviewer #1: Yes

Reviewer #2: Partly

2. Has the statistical analysis been performed appropriately and rigorously?

Reviewer #1: N/A

Reviewer #2: Yes

3. Have the authors made all data underlying the findings in their manuscript fully available?

Reviewer #1: No

Reviewer #2: Yes

4. Is the manuscript presented in an intelligible fashion and written in standard English?

Reviewer #1: Yes

Reviewer #2: Yes

Reviewer #1: Dear Author

I have thoroughly reviewed your paper titled (Performance management and development system in South Africa, a necessary evil. Qualitative study) and I would like to commend you on the valuable insights provided. The study brings important perspectives on the implementation of the PMDS in the healthcare sector, particularly in the PHC setting. However, I believe the paper would benefit from some revisions before publication. Specifically, there are several areas that require further clarification and enhancement.

Thank you for your hard work and dedication to this important topic.

Requerment:

abstract:

- The sentence (Accomplishing performance management’s objectives are critical in the current global challenges of the shortage of healthcare workers and providing equitable, quality healthcare for all.) could be restructured for better readability. Consider rephrasing to (Accomplishing the objectives of performance management is critical in addressing the global challenges of healthcare worker shortages and ensuring equitable, quality healthcare for all)

- The methodology description lacks details on participant selection criteria and sample size. While it mentions (Stratified purposive sampling was used) specifying the number of focus groups, participants, and justification for stratification would strengthen the methodological transparency.Optional)

the introduction section:

The sentence (PM is a means to an end if organisational performance links to individual staff performance) is somewhat ambiguous. Please consider rephrasing it for clarity, such as (PM is effective only when organisational performance is directly linked to individual staff performance)

The section mentions (Despite the gains of PHC, such as access to health and equity, PHC performance in SA could be better) I see this is a broad statement. And I think that the providing of more specific data or references to illustrate PHC's shortcomings in South Africa would strengthen the argument.

the problem statement section:

The sentence (one could question if the MPST influences how the doctors based at PHC experience PMDS and thus contribute positively to the DHS) suggests a causal link between MPST and doctors’ experiences with PMDS, but this is not clearly framed as a research question. Rephrasing as (This study seeks to explore whether MPST influences how PHC doctors experience PMDS and its potential impact on the DHS.) I think it would improve clarity and focus.

the study design section:

The sentence (A qualitative interpretive phenomenological research design was appropriate for this study because it is all about searching for meaning in how participants experience the PMDS and accurately presenting their experiences) could be strengthened by briefly explaining why phenomenology, rather than other qualitative approaches (grounded theory and case study), was the best fit for the research objectives.

The phrase (The researcher not only described but interpreted the meaning of the phenomenon) refers to interpretive phenomenology, but it would be useful to briefly mention a specific framework (Heideggerian phenomenology) to ground the approach in established literature.

research setting section:

While the section provides an in-depth explanation of the healthcare system, the connection to the study’s objectives could be clearer. Pleaswe consider adding a sentence at the end explaining how the research setting’s structure and services influence the performance management and development system (PMDS) experiences of medical officers.

Population and Sampling section:

1. While stratified purposeful sampling is mentioned, it would be helpful to explain why grouping participants by employment duration was relevant to the study. Or how does the duration of employment influence their PMDS experiences? I see Adding a brief justification would strengthen this section.

Data Collection section:

Since the researcher is an SFP in the unit, potential power dynamics could have influenced participant responses. While a research assistant was trained, specifying how the researcher mitigated bias during FGDs would strengthen credibility.

The section states that four FGDs were conducted, but it is unclear why this number was chosen. Did it align with theoretical saturation, or was it based on practical considerations? From my point of view I see its important.

Data Analysis section:

While ATLAS.ti is a well-known qualitative analysis tool, briefly explaining why it was chosen over other software (NVivo) could add depth to the methodological justification.

PMDS, a necessary evil

- The quotes from participants provide valuable insights into their experiences. However, it would strengthen the argument to include more varied perspectives, particularly from those who may have a more critical view of PMDS. This would offer a balanced understanding of the system.

- The section introduces the concept of PMDS as a tool for aligning expectations and guiding work in a primary care setting. It would be useful to elaborate more on how PMDS contributes to long-term professional development for doctors, especially those new to primary healthcare.

Benefits

- The section effectively emphasizes the multi-dimensionality of PMDS, but it would be beneficial to further explore how the benefits are perceived differently across varying experience levels of doctors. Or does PMDS have a different impact on junior doctors versus senior ones?

Performance and Learning Culture

- The focus on self-empowerment through reflection is compelling. However, it would be valuable to further explore how the reflection process is structured within PMDS. Is there a formal system in place for doctors to reflect, or is it more informal?

- The section mentions the importance of building collaborations to improve patient care. It would be helpful to include more concrete examples of how such collaborations have led to tangible improvements in patient outcomes, particularly in the context of a multidisciplinary team.

challenges:

1. The idea of involving employees in the process of setting realistic performance assessment targets is important and could be further expanded. It might be helpful to include any existing evidence or studies that support the effectiveness of a bottom-up approach in improving the relevance and fairness of performance assessments in healthcare settings.

Summary of Key Findings

The summary effectively captures the central theme of PMDS being both beneficial and challenging. To strengthen the clarity, it would be helpful to briefly mention the specific benefits that participants highlighted. This can make the reader more aware of the positive aspects that PMDS brings, alongside the challenges.

While the summary touches on challenges such as the lack of capacity and the need for a bottom-up approach, it would benefit from a bit more explanation of the specific context in which these challenges arise (specific sectors or types of employees) to better ground the findings.

Discussion

Can you please mention a specific examples of how PMDS has successfully aligned with strategic goals in other contexts or studies.

The discussion references a managerial PMDS implementation strategy but also highlights gaps in practical application. But I see that the providing concrete examples or at least case studies where these gaps were successfully addressed could strengthen the argument and offer valuable insights for practitioners.

Authentic Leadership

The connection between leadership and PMDS success is well-articulated. To expand on this, you could explore specific leadership behaviors that have proven successful in PMDS implementations. For example, what role does transformational leadership play in fostering engagement and reducing PMDS-related challenges?

The section addresses the need for both hard and soft skills in leadership. It might be useful to provide examples of specific soft skills (emotional intelligence, conflict resolution) that are crucial in overcoming PMDS challenges and fostering a collaborative work environment.

Limitations

1. The use of stratified purposive sampling is noted as a strength, but it may be beneficial to provide more context on how the sample was selected and why these specific participants (MOs in a particular district) were chosen. This would help readers understand the relevance and applicability of the findings to broader populations.

2. Limitations of Generalizability: While the limitations are clearly stated, it could be useful to offer suggestions for future research that could address these limitations. For instance, conducting similar studies in other regions or with a larger and more diverse group of healthcare professionals could increase the study’s external validity.

Implications/Recommendations

1. The recommendation for a fit-for-purpose PMDS implementation tool is important. It would be helpful to elaborate on what (fit-for-purpose) means in this context. For example (what specific features should this tool have to effectively engage both managers and employees?)

Conclusion

1. The conclusion mentions that this is the first study on doctors' PMDS experiences in the setting. This could be emphasized further by discussing why this is significant or what new insights or contributions to the literature does this study make that other studies might not have captured?

for the References section of the paper:

1. The references are cited in various formats, and some URLs appear incomplete (reference #5). It is important to follow a consistent citation style (APA, MLA) throughout the manuscript, ensuring all references are formatted uniformly.

2. Several of the links provided in the references are either outdated or might not lead to active pages anymore (reference #16 and #28). It would be beneficial to verify the URLs before submission to ensure that all resources are accessible to readers.

3. some references (references #9, #18), publication details such as the volume, issue, and page numbers are missing. Including this information will provide clarity and ensure the references are complete and verifiable.

Here are the updated references with complete details to replace the existing ones:

1. Public Service Commission. (2018). Evaluation of the Effectiveness of the Performance Management and Development System for the Public Service. Retrieved from https://www.psc.gov.za/documents/reports/2018/PMDS%20Report%20_%20Final.pdf

2. Mash, R., & Von Pressentin, K. B. (2018). Strengthening the District Health System Through Family Physicians. South African Health Review, 5(4), 33-40. Retrieved from https://journals.co.za/doi/epdf/10.10520/EJC-1449142b2d

3. Public Service Commission. (2022). Chapter 7: Role of Performance Management. Retrieved from https://www.psc.gov.za/conferences/dev-stateconference/Dev%20State%20Papers/Chapter%207%20Perf%20Man.pdf

Reviewer #2: Dear Authors

Thank you for submitting a very interesting paper.

Your research addresses critical gaps in understanding how PMDS influences healthcare delivery and professional growth among healthcare providers. The study employs a robust qualitative, interpretive phenomenological design, utilizing focus group discussions (FGDs) and thematic analysis to extract insights. The overarching theme, PMDS as a "necessary evil," captures the duality of benefits and challenges experienced by healthcare workers.

The strengths of the manuscript are its originality, research design and methodology, clarity of writing and the smooth flow of the paper.

However, the main drawback of the research is the lack of clarity of its purpose, and its relationship with the findings and implications. For example, the problem statement indicates that the PMDS is necessarily evil. However, this is not reflected in the themes as benefits exceed the challenges by far. Also, the discussion on the benefits of PMDS is much longer than the discussion on what can be done to improve PMDS. The findings don’t address the question of why there is a negative discourse as mentioned in the problem statement. Does it mean that the results of the study disprove the negative discourse in literature around PMDS? If so, this needs to be made explicit. Also, the study talks about leadership and governance as focus areas. But it is not clear from the findings how the leaders will govern better based on the results obtained. The discussion seems to focus more on how is PMDS beneficial, rather than what can be done to improve leadership and governance for PMDS. The paper would also benefit from more actionable recommendations for policymakers and healthcare administrators highlighting practical steps for improving PMDS implementation.

Besides, the paper in its current form is very context specific. It would help if the authors can make the study more generalized by referring to what the relevant existing literature conveys, where is the gap, and how the paper is addressing the gap and what are the future directions of research. Deeper integration with existing literature would enhance its impact. Comparing the study’s results with international experiences of performance management systems could provide a more global perspective.

Also, the content of the paper can be restructured so that the factual information doesnt break the flow of the paper. A lot of details, like the process of PMDS assessment and structural information of the target sample of 20 clinics, can go in Annexures or represented through a table or a figure.

**Do you want your identity to be public for this peer review?** For information about this choice, including consent withdrawal, please see our Privacy Policy

Reviewer #1: **Yes: ** Abdulrahman Awadh Aljuaid

Reviewer #2: **Yes: ** Smita Chaudhry

---

## [Author Response · Author response to Decision Letter 1]

7 May 2025

Dear Reviewers,

Thank you for reviewing my manuscript and for your comments. Below are my honest responses.

All implemented corrections have been highlighted in red.

REVIEWER #1: DEAR AUTHOR

I have thoroughly reviewed your paper titled (Performance management and development system in South Africa, a necessary evil. Qualitative study) and I would like to commend you on the valuable insights provided. The study brings important perspectives on the implementation of the PMDS in the healthcare sector, particularly in the PHC setting. However, I believe the paper would benefit from some revisions before publication. Specifically, there are several areas that require further clarification and enhancement.

Thank you for your hard work and dedication to this important topic.

Requirement:

1. Abstract

a) From Reviewer

The sentence (Accomplishing performance management’s objectives are critical in the current global challenges of the shortage of healthcare workers and providing equitable, quality healthcare for all.) could be restructured for better readability.

Consider rephrasing to (Accomplishing the objectives of performance management is critical in addressing the global challenges of healthcare worker shortages and ensuring equitable, quality healthcare for all)

From Author

This has been implemented on page 2 of the revised manuscript.

b) From the Reviewer: The methodology description lacks details on participant selection criteria and sample size. While it mentions (Stratified purposive sampling was used) specifying the number of focus groups, participants, and justification for stratification would strengthen the methodological transparency. Optional)

From the Author

This has been addressed (page 2 of the revised manuscript).

2. The introduction section

a) From the Reviewer

The sentence (PM is a means to an end if organisational performance links to individual staff performance) is somewhat ambiguous.

Please consider rephrasing it for clarity, such as (PM is effective only when organisational performance is directly linked to individual staff performance)

From the Author

This has been addressed (page 3 of the revised manuscript).

b) From the Reviewer

- The section mentions (Despite the gains of PHC, such as access to health and equity, PHC performance in SA could be better) I see this is a broad statement. And I think that the providing of more specific data or references to illustrate PHC's shortcomings in South Africa would strengthen the argument.

From the Author

This has been addressed on pages 4, paragraph 1 of the revised manuscript.

3. From the Reviewer: the problem statement section

- The sentence (one could question if the MPST influences how the doctors based at PHC experience PMDS and thus contribute positively to the DHS) suggests a causal link between MPST and doctors’ experiences with PMDS, but this is not clearly framed as a research question.

Rephrasing as (This study seeks to explore whether MPST influences how PHC doctors experience PMDS and its potential impact on the DHS.) I think it would improve clarity and focus.

From Author

The rephrased sentence has been implemented (page 6 of the revised manuscript)

4. From the Reviewer: the study design section:

i) The sentence (A qualitative interpretive phenomenological research design was appropriate for this study because it is all about searching for meaning in how participants experience the PMDS and accurately presenting their experiences) could be strengthened by briefly explaining why phenomenology, rather than other qualitative approaches (grounded theory and case study), was the best fit for the research objectives.

From Authur

This has been addressed (page 6 of the revised manuscript under the subtitle, “study design”)

From Reviewer

The phrase (The researcher not only described but interpreted the meaning of the phenomenon) refers to interpretive phenomenology, but it would be useful to briefly mention a specific framework (Heideggerian phenomenology) to ground the approach in established literature.

From Author

This section has been revised (page 7 of the revised manuscript under the subtitle, “study design”)

5. From Reviewer: research setting section

- While the section provides an in-depth explanation of the healthcare system, the connection to the study’s objectives could be clearer. Please consider adding a sentence at the end explaining how the research setting’s structure and services influence the performance management and development system (PMDS) experiences of medical officers.

From Author

The section on research setting has been revised taken into consideration the second reviewers comments (first paragraph, page 7 & 8 of the manuscript).

From Reviewer: Population and Sampling section

1. While stratified purposeful sampling is mentioned, it would be helpful to explain why grouping participants by employment duration was relevant to the study. Or how does the duration of employment influence their PMDS experiences? I see Adding a brief justification would strengthen this section.

From Author

This section has been revised for clarity (page 9, subtitle “population and sampling” of the revised manuscript).

6. From Reviewer: Data Collection section:

- Since the researcher is an SFP in the unit, potential power dynamics could have influenced participant responses. While a research assistant was trained, specifying how the researcher mitigated bias during FGDs would strengthen credibility.

From Author

The author added a sentence (page 10, paragraph 1) and paragraph 2, page 12 under reflexivity.

From Reviewer: The section states that four FGDs were conducted, but it is unclear why this number was chosen. Did it align with theoretical saturation, or was it based on practical considerations? From my point of view I see its important.

From Author

This has been clarified under the subtitles “population and sampling” and “data collection” (pages 9 of the revised manuscript).

7. From Reviewer: Data Analysis section:

- While ATLAS.ti is a well-known qualitative analysis tool, briefly explaining why it was chosen over other software (NVivo) could add depth to the methodological justification.

From Author

This has been addressed under “data analysis (page 10 of the revised manuscript).

8. From Reviewer: PMDS, a necessary evil

-a) The quotes from participants provide valuable insights into their experiences. However, it would strengthen the argument to include more varied perspectives, particularly from those who may have a more critical view of PMDS. This would offer a balanced understanding of the system.

From Author

The critical views around the discourse on PMDS by participants were grouped under the theme “challenges.” Actually, it was surprising to the researcher that participants had numerous positive experiences (from page 38 of the manuscript)

b) From Reviewer: The section introduces the concept of PMDS as a tool for aligning expectations and guiding work in a primary care setting. It would be useful to elaborate more on how PMDS contributes to long-term professional development for doctors, especially those new to primary healthcare.

From Author

This has been addressed (page 16, paragraph 1 of the revised manuscript).

9. From reviewer: Benefits

a) The section effectively emphasizes the multi-dimensionality of PMDS, but it would be beneficial to further explore how the benefits are perceived differently across varying experience levels of doctors. Or does PMDS have a different impact on junior doctors versus senior ones?

From author

This has been addressed by adding the graphical analysis of the contributions from the focus group discussions (page 15 of the revised manuscript).

b) Performance and Learning Culture

From Reviewer: The focus on self-empowerment through reflection is compelling. However, it would be valuable to further explore how the reflection process is structured within PMDS. Is there a formal system in place for doctors to reflect, or is it more informal?

From Author

Reflection is informal. This has been revised for clarity (page 19 of the manuscript).

d) From Reviewer

The section mentions the importance of building collaborations to improve patient care. It would be helpful to include more concrete examples of how such collaborations have led to tangible improvements in patient outcomes, particularly in the context of a multidisciplinary team.

From Author

A relevant quote has been added (page 21 of the manuscript).

10. From Reviewer: Challenges:

1. The idea of involving employees in the process of setting realistic performance assessment targets is important and could be further expanded. It might be helpful to include any existing evidence or studies that support the effectiveness of a bottom-up approach in improving the relevance and fairness of performance assessments in healthcare settings.

From Author

This evidence has been added (page 39 of the manuscript under challenges).

11. From Reviewer: Summary of Key Findings

a) The summary effectively captures the central theme of PMDS being both beneficial and challenging. To strengthen the clarity, it would be helpful to briefly mention the specific benefits that participants highlighted. This can make the reader more aware of the positive aspects that PMDS brings, alongside the challenges.

From Author

This section has been revised (page 38 of the revised manuscript under ‘summary of key findings’.

b) From Reviewer

- While the summary touches on challenges such as the lack of capacity and the need for a bottom-up approach, it would benefit from a bit more explanation of the specific context in which these challenges arise (specific sectors or types of employees) to better ground the findings.

From Author

This has been added to the summary (page 38 of the revised manuscript).

12. From Reviewer: Discussion

a) Can you please mention a specific examples of how PMDS has successfully aligned with strategic goals in other contexts or studies.

From Author

This has been added (page 38 of the revised manuscript under discussion).

b) From Reviewer

The discussion references a managerial PMDS implementation strategy but also highlights gaps in practical application. But I see that the providing concrete examples or at least case studies where these gaps were successfully addressed could strengthen the argument and offer valuable insights for practitioners.

From Author

This has been addressed on page 39, last paragraph of the revised manuscript.

c) From Reviewer: Authentic Leadership

The connection between leadership and PMDS success is well-articulated. To expand on this, you could explore specific leadership behaviors that have proven successful in PMDS implementations. For example, what role does transformational leadership play in fostering engagement and reducing PMDS-related challenges?

From Author

This information has been added (page 43 of the revised manuscript under “Authentic leadership)”

d) From Reviewer

The section addresses the need for both hard and soft skills in leadership. It might be useful to provide examples of specific soft skills (emotional intelligence, conflict resolution) that are crucial in overcoming PMDS challenges and fostering a collaborative work environment.

From Author

This has been addressed on page 43 of the revised manuscript under “authentic leadership”, last paragraph.

13. From Reviewer: Limitations

a) The use of stratified purposive sampling is noted as a strength, but it may be beneficial to provide more context on how the sample was selected and why these specific participants (MOs in a particular district) were chosen. This would help readers understand the relevance and applicability of the findings to broader populations.

From Author

The reason for stratified sampling has been clarified in the methodology section.

b) From Reviewer: Limitations of Generalizability: While the limitations are clearly stated, it could be useful to offer suggestions for future research that could address these limitations. For instance, conducting similar studies in other regions or with a larger and more diverse group of healthcare professionals could increase the study’s external validity.

From Author

This has been revised (pages 46 and 47 of the manuscript under strengths and limitations’)

14. From Reviewer: Implications/Recommendations

a) The recommendation for a fit-for-purpose PMDS implementation tool is important. It would be helpful to elaborate on what (fit-for-purpose) means in this context. For example (what specific features should this tool have to effectively engage both managers and employees?)

From Author

This has been addressed (page 47 of the manuscript under ‘implications and recommendations).

15. From Reviewer: Conclusion

1. The conclusion mentions that this is the first study on doctors' PMDS experiences in the setting. This could be emphasized further by discussing why this is significant or what new insights or contributions to the literature does this study make that other studies might not have captured?

From Author

This has been revised. Page 47.

16. From Reviewer: for the References section of the paper

1. The references are cited in various formats, and some URLs appear incomplete (reference #5). It is important to follow a consistent citation style (APA, MLA) throughout the manuscript, ensuring all references are formatted uniformly.

2. Several of the links provided in the references are either outdated or might not lead to active pages anymore (reference #16 and #28). It would be beneficial to verify the URLs before submission to ensure that all resources are accessible to readers.

3. some references (references #9, #18), publication details such as the volume, issue, and page numbers are missing. Including this information will provide clarity and ensure the references are complete and verifiable.

Here are the updated references with complete details to replace the existing ones:

1. Public Service Commission. (2018). Evaluation of the Effectiveness of the Performance Management and Development System for the Public Service. Retrieved from https://www.psc.gov.za/documents/reports/2018/PMDS%20Report%20_%20Final.pdf

2. Mash, R., & Von Pressentin, K. B. (2018). Strengthening the District Health System Through Family Physicians. South African Health Review, 5(4), 33-40. Retrieved from https://journals.co.za/doi/epdf/10.10520/EJC-1449142b2d

3. Public Service Commission. (2022). Chapter 7: Role of Performance Management. Retrieved from https://www.psc.gov.za/conferences/dev-stateconference/Dev%20State%20Papers/Chapter%207%20Perf%20Man.pdf

Author

The reference section has been revised from page 49 in keeping with NLM/ICMJE style as indicated by the Plos One journal.

Reviewer #2: Dear Authors

Thank you for submitting a very interesting paper.

Your research addresses critical gaps in understanding how PMDS influences healthcare delivery and professional growth among healthcare providers. The study employs a robust qualitative, interpretive phenomenological design, utilizing focus group discussions (FGDs) and thematic analysis to extract insights. The overarching theme, PMDS as a "necessary evil," captures the duality of benefits and challenges experienced by healthcare workers.

The strengths of the manuscript are its originality, research design and methodology, clarity of writing and the smooth flow of the paper.

1. From Reviewer

However, the main drawback of the research is the lack of clarity of its purpose, and its relationship with the findings and implications. For example, the problem statement indicates that the PMDS is necessarily evil. However, this is not reflected in the themes as benefits exceed the challenges by far.

From Author

The purpose of the study has been clarified (page 6, paragraph 2 under “problem statement”).

PMDS, a necessary evil, in the introductory section under discussion, has been revised for clarity (page 16 of the manuscript).

2. From Reviewer

Also, the discussion on the benefits of PMDS is much longer than the discussion on what can be done to improve PMDS.

---

## [Editor Report · Decision Letter 1]

Performance management and development system in South Africa, a necessary evil. Qualitative study

PONE-D-24-57158R1

Dear Dr. OZOEMENA Joan IBEZIAKO

We’re pleased to inform you that your manuscript has been judged scientifically suitable for publication and will be formally accepted for publication once it meets all outstanding technical requirements.

Kind regards,

Margaret Williams, Ph.D

Academic Editor

PLOS ONE
---

## [Editor Report · Acceptance letter]

PONE-D-24-57158R1

PLOS ONE

Dear Dr. Ibeziako,

I'm pleased to inform you that your manuscript has been deemed suitable for publication in PLOS ONE. Congratulations! Your manuscript is now being handed over to our production team.

Kind regards,

on behalf of

Professor Margaret Williams

Academic Editor

PLOS ONE